# Analysis of the Waiting Time in Clinic Registration of Patients with Appointments and Random Walk-Ins

**DOI:** 10.3390/ijerph20032635

**Published:** 2023-02-01

**Authors:** Jin Kyung Kwak

**Affiliations:** Business Administration, Ewha Womans University, Seoul 03760, Republic of Korea; jkkwak@ewha.ac.kr

**Keywords:** healthcare, waiting of patients, queues with different priorities, pooling, clinic registration, appointments, walk-in

## Abstract

Healthcare institutions generally use an appointment system. However, patients often need to receive medical services unexpectedly. If they visit a clinic without an appointment, they may have to wait for a long time, as their priority is low. In this study, we investigated whether the clinic registration system can be improved by separating the queues and resources for different types of patients. From our simulation results, we found that under a certain setup, the separation policy does not effectively reduce the walk-ins’ waiting time, nor improve the service. The study gives valuable managerial insights into the factors affecting patients’ waiting times. As the number of random walk-ins is relatively higher, the service times are longer, and the no-show rate of appointments is lower, separation may reduce the waiting time of walk-in patients.

## 1. Introduction

The global aging population and the recent outbreak of COVID-19 have caused the demand for healthcare services to be on a sharp rise unlike anything seen before. Hospitals need to respond to patients’ requests effectively and efficiently. One of many factors affecting patients’ satisfaction is their waiting time before receiving health services [1,2,3]. Long waiting times have been recognized as a major complaint from patients and many existing studies have suggested reducing the waiting duration as a solution.

The major related research stream is about appointment scheduling for hospitals [4,5,6,7]. Appointment systems have been widely used in healthcare institutions to offer a timely medical service by reducing arrival variability. Previous studies on reducing waiting times in hospitals focused on improving appointment systems. Recently, several extensions have been explored to incorporate the inherently complicated characteristics of medical services. These include: assuming service interruptions such as emergencies [8]; considering no-shows, walk-ins, variability of service time and cost of doctors’ time to patients’ time [9]; situations involving unpunctual patients, multiple servers, and no-shows [10]; an adaptive appointment system that adjusts dynamically [11]; scheduling for routine and same-day patients when some patients revisit on the same day [12]; offering medical service in two stages, e.g., from nurses to doctors [13];, and allocating different time-slot durations [14].

Unlike the aforementioned literature, which mostly pursued better appointment scheduling in hospitals, our study focuses more on the situation where many patients walk in randomly without appointments. An unexpected demand may occur very commonly in the healthcare industry because patients’ sudden illnesses cannot be predicted in advance for appointments. Some previous studies have considered the presence of walk-in patients [15,16]; however, they focused on appointment scheduling. Wang et al. [15] demonstrated that appointment scheduling with walk-ins should be different from that without. Robinson and Chen [16] compared traditional and open-access policies for appointment scheduling when there is one doctor and the service time is deterministic.

The present study is motivated by the need to accommodate random walk-in patients. Numerous healthcare institutions use appointment systems for patients’ convenience; however, in addition to regular checkups, patients need to visit hospitals or clinics when they are unexpectedly ill. These patients may be allowed to visit the clinics without appointments, but they may have to wait for a very long time because other patients with appointments have a higher priority. On the other hand, these high-priority patients may not show up at the reserved time as they may have time conflicts or simply forget the appointment. The existing literature on healthcare appointment scheduling has addressed the no-shows of patients [4,8,9,10,11,17].

The critical research question in the present study is: what if we add a queue for walk-in patients separate from the patients with appointments? Does this separation help to solve the issues of the waiting time of walk-ins and no-shows of patients with appointments? Under what circumstances is it more helpful?

To pool or not to pool the queues has been a widely studied research question, and many existing studies in the literature have addressed the issues of the pooling or separation strategies in queueing systems [18,19,20,21]. These studies mostly focused on the circumstances where pooling the queues is not always beneficial, especially when there is jockeying [18], a queueing network with flexible servers [19], an emergency department [20] or different call types in a call center [21]. The major difference with this study is that it is for an outpatient clinic allowing both appointments and random walk-ins, while giving a higher priority to patients with appointments.

The most common performance measures of previous studies were the waiting time of patients, and the idle time and overtime of doctors [9,10,12,16,22]. The present study equally addresses the waiting time of patients; in particular, we track the waiting time separately depending on the patient type with different priorities. Another measure considered is the total number of patients served by doctors, as this determines the level of service or revenue.

By conducting a simulation study using the ARENA software (version 16, Rockwell Automation, Inc., Milwaukee, WI, USA), we compare the measures in a pooled system (having one queue with priority given to appointments) and those in a separated system (having a separate queue and doctors for random walk-in patients). The simulation results showed that under a certain setup, separating queues for walk-ins does not effectively reduce their waiting time, and even increases the waiting time of patients with appointments. Separation may be helpful when there are relatively more walk-in patients, the time of medical service is longer, and/or the no-show probability of appointments is lower.

The subsequent parts of the study are structured as follows: Section 2 introduces the models and methods used in this study; Section 3 analyzes the simulation results; Section 4 discusses the managerial implications; and finally, Section 5 concludes the paper with the limitations and future research directions.

## 2. Models and Methods

We developed two simulation models describing a healthcare system serving two types of patients—patients who make an appointment in advance, and patients who visit the clinics without appointments. Throughout the paper, we designate the first type as “patients with appointments” and the second type as “walk-ins”. Figure 1 illustrates a clinic registration system where patients with advanced appointments have a higher priority than walk-ins to receive medical service. Walk-ins can be served only when there is no patient with an appointment in the queue. We denote this model as Model_P as the queue is pooled for both types of patients.

Figure 2 shows Model_S, where there are separate queues and separate resources of medical service (doctors) for each type of patient. Both Model_P and Model_S are simulation models for an outpatient clinic.

We assumed that for both models, the inter-arrival time distribution followed a normal distribution for patients with appointments and an exponential distribution for walk-ins. Considering that the appointment time slots usually have fixed intervals, the assumption of a normal distribution seems reasonable. Random arrivals are commonly modeled with an exponential distribution of inter-arrival times.

This study compares the outcomes of the two models—the total number of patients who were served and the waiting time of each type of patient—to obtain managerial insights on clinic registration systems. We performed a simulation using the ARENA software, which is good for analyzing queueing problems. Once the arrival of patients with appointments was generated, we incorporated the no-show probability. The replication length was set to 240 min, which generally covers the morning (9:00–13:00) and afternoon (14:00–18:00) time slots in a clinic.

## 3. Analysis

To analyze which factors affect patient waiting, we simulated and compared some scenarios (Table 1). The inter-arrival time is the time between the arrivals of patients and the service time is the duration of medical service to each patient provided by a doctor. The no-show rate is the probability that the patients with appointments do not show up at the reserved time. Under our simulation setup, we assigned one doctor for 12 patients per hour. For example, if the inter-arrival time distribution of walk-ins is exponential with a mean of 5 min and that of patients with appointments is normal with a mean of 5 ± 2 min, there are two doctors with one queue in Model_P and one doctor with one queue for each type of patient in Model_S. The choice of parameters does not affect the managerial insights when comparing the two systems because we analyzed the relationship between the factors of interest and waiting-related measures, though the simulation setup was designed to have a high utilization of resources for facilitating patient waiting. Some previous studies on healthcare queueing systems provided information on patient arrivals, but they varied over a wide range: 30 per day [23], 300 per day [24], or different in time and about 15 per hour [25]. An arrival rate of 12 patients per hour seems reasonable within the ranges in the scientific literature.

### 3.1. The Pooling Effect

From the results yielded by the ARENA software, we obtained information on the “Total Number Seized”, which indicates the total number of patients treated by the doctors. This measure is directly linked to revenue because some of the patients who have waited too long in the queue might leave without service, leading to lost sales. A few existing studies [26,27] mentioned an acceptable threshold for waiting time to be about 30 min. The present study did not specify an acceptable threshold. However, the measure “Total Number Seized” gives a good approximation of the number of patients served, assuming the patients who arrive but do not receive service at the end of the simulation might leave the system due to the long waiting time. Even if they do not leave, the doctors would have to work overtime to serve those patients, incurring higher costs.

The simulation results show that in most cases (except for the cases where the number of walk-ins is two times that of patients with appointments and the no-show rates are 0.1 and 0.2, respectively), Model_P serves more patients than Model_S. Although walk-ins may have to wait for a very long time before receiving service, having one queue with priority given to appointments serves more patients than having separate queues to different doctors. This result can be explained by the concept of pooling, which prevents the situation where one resource is idle while the other resource is busy.

In the present study, the “pooling effect” is defined as a measure of service received indicating how much more beneficial pooling the queue is for the two types of patients rather than separating the queues by serving more patients. The pooling effect was computed by Equation (1) where *N_P_* is the total number of patients served in Model_P and *N_S_* is the total number of patients served in Model_S.
(1)$$\mathrm{The}\;\mathrm{pooling}\;\mathrm{effect}=\frac{N_{P}-N_{S}}{N_{S}}$$

The simulation data show that the pooling effect ranges from −0.0038 to 0.1691 with an average of 0.0370. To analyze the effect of the factors, we computed the average values of the pooling effect for different no-show rates under the scenarios classified in Table 1.

The upper left quadrant of Figure 3 describes the relationship between the average value of the pooling effect and the ratio of the number of walk-ins to the number of patients with appointments. As the ratio increased, the pooling effect tended to decrease. This result implies that if there are relatively more walk-ins, having separate queues will not be so detrimental to the service level.

The upper right quadrant of Figure 3 illustrates the average value of the pooling effect with respect to the mean service time. Since the service time provided by doctors has a higher mean, the pooling effect is lower. Having just one queue serves more patients when the service time is shorter.

The lower left quadrant of Figure 3 shows the impact of arrival variability of patients with appointments on the pooling effect. There is no significant trend in their relationship. Arrival variability does not seem to affect whether the queues are pooled.

The lower right quadrant of Figure 3 shows that the average value of the pooling effect increases as the no-show rate of patients with appointments increases. As the number of patients not showing up despite making appointments in advance increases, having one queue becomes preferable to serve more patients in total. The magnitude of the pooling benefit equally increases with the no-show rate.

### 3.2. Time Saving and Loss from Separation

It is expected that having a separate registration waiting line for random walk-in patients will help reduce their waiting time. Conversely, with the same number of resources (doctors), the waiting time for patients with appointments will increase compared to when there is only one queue, where they have a higher priority to be served. In the present study, we define the two measures explaining the time saving of walk-ins and the time loss of patients with appointments. 

“Saving” is defined in Equation (2) where *WT_P_* is the walk-ins’ waiting time in Model_P and *WT_S_* is the walk-ins’ waiting time in Model_S. Therefore, this measure indicates how much time can be saved for walk-ins if we separate the queues, rather than having only one queue for clinic registration. Under our simulation setup, the time saved ranged from −1.4183 to 0.4191 with an average of 0.0011. The measure sometimes has a negative value, meaning that the walk-ins’ waiting time can be even longer when they have a registration system separate from the pooled system. In particular, when the no-show rate is high, pooling the queues is more efficient; otherwise, the queue of walk-ins will be too crowded:(2)$$\mathrm{Saving}=\frac{WT_{P}-WT_{S}}{WT_{P}}$$

“Loss” is defined in Equation (3), where *AT_P_* is the waiting time of the patients with appointments in Model_P and *AT_S_* is the waiting time of the patients with appointments in Model_S. It describes how much more time the patients with appointments lose due to separation instead of the pooled system as a rate. Under the simulation setup, Loss ranges from 0.2247 to 15.7730 with an average of 4.9511. Compared to the walk-ins’ time saving from separation, the time loss rate of the patients with appointments was much larger in our scenarios:(3)$$\mathrm{Loss}=\frac{AT_{S}-AT_{P}}{AT_{P}}$$

To analyze the effect of factors of interest described in Table 1, we computed average values of Saving and Loss for different no-show rates with respect to each factor as in Figure 4. The graph with circle-shaped dots (blue line) shows Saving in the left axis while the graph with triangle-shaped dots (orange line) shows Loss in the right axis.

The upper left quadrant of Figure 4 shows the Saving and Loss with respect to the ratio of the number of walk-ins to the number of patients with appointments. As the ratio increases, walk-ins’ time saving from separation increases whereas the time loss of patients with appointments also increases. When there is a relatively lower number of walk-ins, even walk-in patients do not save time by separating queues in the registration system.

The upper right quadrant of Figure 4 depicts the increasing trend of both Saving and Loss with respect to mean service time. As the medical service takes longer, the walk-ins can save more time by separation, but patients with appointments lose more time. When the service time is relatively short, even walk-in patients do not save time by separating queues.

The lower left quadrant of Figure 4 implies no significant trend of Saving or Loss with respect to the arrival variability of patients with appointments.

The lower right quadrant of Figure 4 shows that the average values of Saving and Loss both decrease as the no-show rate increases. The separation benefits the walk-ins when the no-show rate is relatively lower. When no-show rates are relatively high, separation is not beneficial even to the walk-ins.

## 4. Discussion

When there are unexpected, urgent needs for medical services, healthcare clinics may allow walk-in patients without appointments. The problem is, since their priority is lower than that of patients with appointments, walk-ins may have to wait for a very long time before receiving service. This study stemmed from the following research question: what if the system has separate queues for each type of patient? Separation is expected to reduce the waiting time of walk-ins.

The results of this simulation study, however, show that separating the queues and resources for each type of patient does not improve the clinic registration system. Both patient types sharing the queue clearly results in more patients served when compared to the separation policy, because of the so-called pooling effect. Separating the queues is expected to reduce the waiting time of walk-ins but the time savings are minimal compared to the significant time loss of patients with appointments.

Many previous studies [18,19,20,21] have raised the research question of whether or not to pool the queues in a certain setting, and have argued that pooling is not always beneficial. However, the results of this study show that pooling can be more beneficial when there are two types of customers with different priorities, and the higher priority is given to the patients with appointments in a healthcare context.

Nevertheless, we acquired valuable practical implications. If there are more walk-in patients, the service time is longer, and the no-show rate of the patients with appointments is lower, the separate queue may be helpful. These conditions help make decisions regarding the type of clinic registration system to design.

The need for urgent medical services in the analysis of patient waiting time has also been addressed in previous studies [16,28], although their results do not seem consistent with what we found in the simulation study. The open-access policy in the study of Robinson and Chen [16] significantly outperforms the traditional schedule in most cases because the patients who make a same-day appointment in the morning can alternate with the no-shows. Since the open-access policy allows the patients to make an appointment for the same day, they have the same queue as the existing patients who made appointments in advance. If clinics allow same-day appointments for walk-ins, it may reduce their waiting time in the system, but they may have to receive medical service later than when they walk in. In a study of a radiotherapy outpatient department [21], the separation of queues had a better performance than pooling as two or more different service types were involved. Their customer types were different from those in the present study in that there were no priority orders among the types. The waiting situation in hospitals can vary depending on the characteristics of the clinic, thereby varying optimal policies for registration.

The results and implications of the present study apply to any service system that has customers with different priorities, where the customers who make appointments in advance receive a higher priority. Pooling the queues and resources is beneficial in general, but under certain circumstances, separation can help reduce patient waiting time.

## 5. Conclusions

With an increasing demand for healthcare services, patients’ requests need to be responded to as well. Since waiting time is one of the most crucial components of patient satisfaction, designing the healthcare registration system in a way that reduces patient waiting time is a valuable subject both in practice and in academia. There is much literature on this topic, but most studies focus on appointment scheduling while the present study is motivated by the fact that health issues occur suddenly in most cases, so demands exist without appointments.

In the present study, we simulated two models (the pooled vs. separated system) and compared the differences using the ARENA software. We found that the pooled system serves more patients than the separated system due to the so-called pooling effect, which prevents the case where some resources are idle while other resources are busy. The results also show that the time saving of walk-ins is relatively minimal while the time loss of patients with appointments is significant.

We define several performance measures—the “pooling effect” indicating how many more patients are served by pooling than by separation; “Saving”, indicating how much more waiting time is saved for walk-ins; and “Loss”, indicating how much more time is lost for the patients with appointments. The pooling effect increases as the ratio of the number of walk-ins to the number of patients with appointments decreases, as the mean service time decreases, and as the no-show rate increases. Saving and Loss increase as the ratio of the number of walk-ins to the number of patients with appointments increases, as the mean service time increases, and as the no-show rate decreases. The arrival variability of walk-ins does not affect any outcome.

The actual parameter values will be different depending on each clinic’s situation, albeit we can have a better understanding of the effect of factors on the waiting times of patients with different priorities. Although a simplification of models helps to provide clear insights, there exist limitations regarding detailed circumstances. For example, we only considered the cases involving a high utilization of doctors, because we aimed to observe the results when patients’ waiting occurred and there may have been lost sales. By incorporating the complicated nature of medical service into the models, we can obtain more practical implications. The present study is expected to act as a starting point for future studies.

Will the results of the present study be different for a two-stage system where the medical service is provided in sequence, such as first receiving a nurse consultation and then a doctor checkup? We may also consider that there are more than two priorities in categorizing the patient types, e.g., emergent patients should receive an even higher priority than the patients with appointments. Will having all these patients make any difference in waiting time? These extensions can be explored based on the present study, and the ultimate decision of pooling or not pooling in clinic registration can be made for a specific environment.

## Figures and Tables

**Figure 1 ijerph-20-02635-f001:**
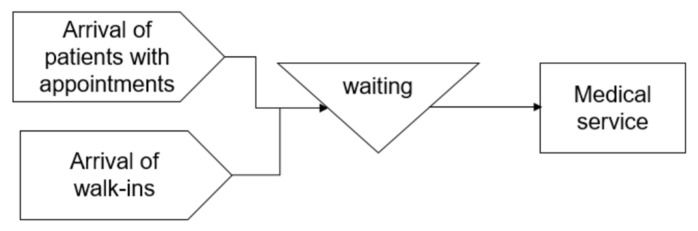
Model_P.

**Figure 2 ijerph-20-02635-f002:**
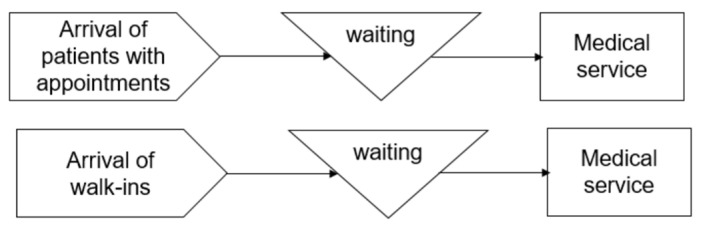
Model_S.

**Figure 3 ijerph-20-02635-f003:**
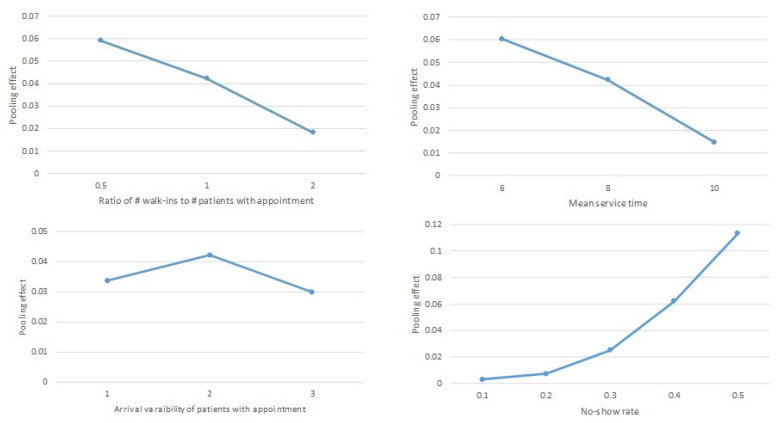
The pooling effect with respect to parameters.

**Figure 4 ijerph-20-02635-f004:**
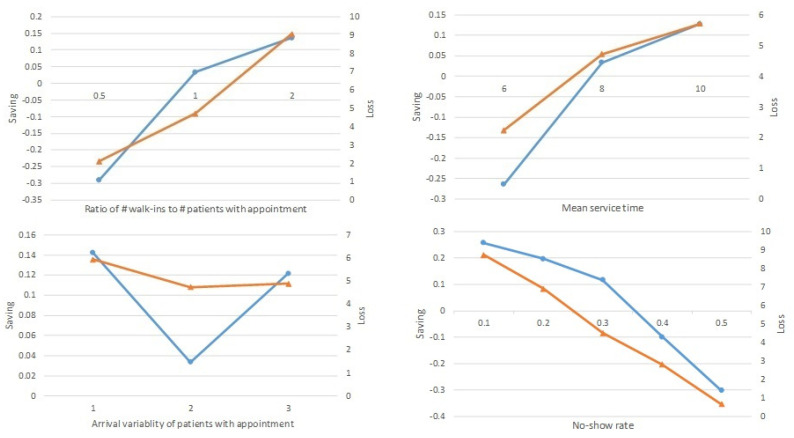
Saving and Loss with respect to parameters.

**Table 1 ijerph-20-02635-t001:** Scenarios in the simulation study.

Factor of Interest	Parameter Change	Other Settings
Ratio of # of walk-ins to # of patients with appointments	Inter-arrival time distribution of [walk-ins, patients with appointments]: (Expo(5), Norm(2.5,2)) (Expo(5), Norm(5,2)) (Expo(2.5), Norm(5,2))	Service time distribution Norm(8,2) No-show rate {0.1,0.2,0.3,0.4,0.5} # doctors 3 (1 and 2 for S ^1^) # doctors 2 (1 and 1 for S) # doctors 3 (2 and 1 for S)
Service time distribution	Norm(6,2) Norm(8,2) Norm(10,2)	Inter-arrival time distribution of walk-ins: Expo(5) patients with appointments: Norm(5,2) No-show rate {0.1,0.2,0.3,0.4,0.5} # doctors 2 (1 and 1 for S)
Inter-arrival time variability of patients with appointments	Norm(5,1) Norm(5,2) Norm(5,3)	Inter-arrival time distribution of walk-ins: Expo(5) No-show rate {0.1,0.2,0.3,0.4,0.5} Service time distribution Norm(8,2) # doctors 2 (1 and 1 for S)
No-show rate	0.1, 0.2, 0.3, 0.4, 0.5	Inter-arrival time distribution of walk-ins: Expo(5) patients with appointments: Norm(5,2) Service time distribution Norm(8,2) # doctors 2 (1 and 1 for S)

^1^ Allocating doctors for Model_S (1 for walk-ins and 2 for patients with appointments).

## Data Availability

Data can be provided upon request to the author.

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
