# Peer review of "Analysis of the Waiting Time in Clinic Registration of Patients with Appointments and Random Walk-Ins"

_ijerph, 2023, doi:10.3390/ijerph20032635_

Round 1

Reviewer 1 Report

I understand that this manuscript provides a comparative analysis of the effect of pooling and separation policies on the waiting time of patients with appointments and random walk-ins in medical services. Research questions are clearly presented, making it easy to read through the manuscript. However, it does not rise up to the publication standard of this technical journal for the following reasons:
1. The contribution of the manuscript is not very clear. To pool or not to pool the queues is a question that has been widely studied for a long time. Although many of the relevant studies are not conducted in the context of healthcare operations, their findings may also be applicable in this field. A clear gap between these previous articles and this manuscript needs to be identified in the introduction section and incorporated into the research questions and experiments.

2. Although a brief literature review on appointment scheduling is presented in the introduction section, the one about pooling and separation strategies in queuing systems is missing. There are many excellent papers in this area, such as:
[1] Rothkopf M H, Rech P. Perspectives on queues: Combining queues is not always beneficial[J]. Operations Research, 1987, 35(6): 906-909.

[2] Mandelbaum A, Reiman M I. On pooling in queueing networks[J]. Management Science, 1998, 44(7): 971-981.

[3] van Dijk N M, van der Sluis E. To pool or not to pool in call centers[J]. Production and Operations Management, 2008, 17(3): 296-305.

[4] Song H, Tucker A L, Murrell K L. The diseconomies of queue pooling: An empirical investigation of emergency department length of stay[J]. Management Science, 2015, 61(12): 3032-3053.

3. Although considerable comparative experiments on the two models are conducted in this research, the analysis of the experimental results is not adequate. Neither the analysis nor the discussion chapters provide sufficiently profound insights. It seems that all the results are relevant to the rationale of load balancing when pooling the two queues.

Reviewer 2 Report

Dear Author,

the paper “Analysis of the waiting time in clinic registration of patients with appointments and random walk-ins” is well written and clearly understandable. However, there are some issues that should be addressed:

1) The paper lacks of the description of the context in which the model has been set-up. There is no information on the type of healthcare system included in the model, if it is a “simulated” healthcare system or if it has been used the healthcare system of a certain country. This also affects the data included in the model: as an example, it is not clear on which basis the author included a total of 12 patients per doctor per hour in the model. I suggest to explain more how the model was constructed, and on which (scientific) basis. Does the model refer to outpatients visits, or only to GP visits? Is the model generalizable to different healthcare systems?

2) Moreover, the discussion section needs improvement: the results of the paper needs to be compared with scientific literature from all over the word. I suggest to improve the discussion section and to add further comparisons between the results of the study and the scientific literature..

3) In figures 7, 8, 9 and 10 it is not clear what the orange and blue lines represent, as there is no explanation in the figure and in the caption. I also suggest combining the four figures in one figure with four graphs. The same suggestion applies to figures 3, 4, 5 and 6.

Round 2

Reviewer 1 Report

The revised version clearly states the novelty and significance of this study. It is satisfactory enough.